# Equity considerations in outcome measures of the HIV pre-exposure prophylaxis care continuum in high-income countries: a systematic review protocol

Melissa Cabecinha ,[1] Danielle Solomon,[2] Greta Rait,[1,3] John Saunders,[3,4] Hamish Mohammed,[5,6] Lorraine Katherine McDonagh[1,3]

► Prepublication history and additional materials for this paper is available online. To view these files, please visit the journal online (http://dx.doi.org/10.1136/bmjopen-2020-040701).

**Correspondence to**
Melissa Cabecinha;
m.cabecinha@ucl.ac.uk

## ABSTRACT

**Introduction** HIV pre-exposure prophylaxis (PrEP) is an effective intervention to reduce acquisition of HIV. PrEP provision has increased in recent years, however, it is not known whether PrEP implementation has been equitably implemented across all risk groups, particularly groups experiencing high levels of health inequity. A PrEP care continuum (PCC) has been proposed to evaluate the success of PrEP implementation programmes, but the extent to which health equity characteristics are currently taken into account in the PCC has not been described. The objectives of this proposed systematic review are to (i) identify and collate outcome measure definitions for the main stages of the PCC (awareness, acceptability, uptake, adherence and retention), (ii) describe how equity characteristics are considered in outcome definitions of the PCC and (iii) describe data sources for capturing equity characteristics.

**Methods and analysis** Quantitative studies published between 1 January 2012 and 3 March 2020 will be included. Five databases (MEDLINE, PubMed, Embase, Cumulative Index to Nursing and Allied Health Literature, Applied Social Sciences Index and Abstracts) will be searched to identify English language publications that include an outcome measure definition of at least one of the five main stages of the PCC. Risk of bias will be assessed using the Effective Public Health Practice Project Quality Assessment Tool for Quantitative Studies. Data on outcome measure definitions and equity characteristics will be extracted. Results will be presented in a narrative synthesis and all findings will be reported in accordance with the Preferred Reporting Items for Systematic Reviews and Meta-Analyses guidelines.

**Ethics and dissemination** Ethical approval is not required. The results will be disseminated via submission for publication to a peer-reviewed journal when complete. The review findings will have relevance to healthcare professionals, policymakers and commissioners in informing how to best evaluate PrEP implementation programmes and inform new implementation strategies for vulnerable and less advantaged populations.

**PROSPERO registration number** CRD42020169779.

## Strengths and limitations of this study

► This systematic review will be the first to collate outcome measures for the HIV pre-exposure prophylaxis (PrEP) care continuum from real-world settings and investigate how equity characteristics are considered within existing measures.

► The findings from this review will help inform evaluation and monitoring of PrEP programmes and how to view these evaluations through an equity lens.

► This protocol has been written in accordance with published Preferred Reporting Items for Systematic Review and Meta-Analysis Protocols guidelines, and the review will be reported in accordance with the Preferred Reporting Items for Systematic Reviews and Meta-Analyses statement guidelines and the 2012 Preferred Reporting Items for Systematic Reviews and Meta-Analyses Equity extension.

► This review will not provide specific information on the effectiveness of existing PrEP programmes at reducing health inequity or population vulnerability.

## INTRODUCTION

Oral HIV pre-exposure prophylaxis (HIV PrEP) is a highly effective biomedical intervention for reducing HIV acquisition, and is recommended by the WHO as part of a comprehensive approach to HIV prevention.[1] The efficacy of PrEP has been demonstrated in clinical trials,[2,3] and PrEP provision has been increasing globally in recent years.[4] However, questions remain as to whether real-world implementation strategies are reaching some high-risk and vulnerable populations,[5] particularly those populations experiencing high levels of health inequity.

### Health inequity and HIV

Health inequities are disparities in health status or access to healthcare that are considered unnecessary, avoidable, unfair and

unjust.[6] Health equity is an important consideration in HIV prevention, as HIV disproportionately affects marginalised populations. Gender and sexual minorities,[7–9] ethnic minorities,[7 10] vulnerable populations such as people who inject drugs,[11] people experiencing housing instability[12] and other social disadvantages[13] are at a greater risk of HIV acquisition compared with the general population. These categories are not mutually exclusive; an individual may identify with multiple categories concurrently[14] and the experience of belonging to multiple marginalised groups may contribute to an increased risk of acquiring HIV.[15] HIV prevention programmes, including PrEP provision programmes, must address the needs of marginalised populations and communities in order to have a sustained impact on the decline of HIV transmission.[11]

### Measurable outcomes for PrEP implementation

A PrEP care continuum (PCC) has been proposed[16 17] to evaluate the implementation of PrEP. The main stages of the PCC are as follows: awareness (knowledge of PrEP and indication of relevance), acceptability (whether an individual views PrEP as an acceptable tool for prevention), uptake (engagement in PrEP related care and initiation of PrEP) and adherence (retention in PrEP related care and adherence to treatment). Although the PCC provides a useful framework, in practice the outcomes have been reported with varying definitions. For example, in evaluating the 'uptake' stage of the continuum, a study of veterans in the US reports uptake as the absolute number of people initiating on PrEP within a given time-frame,[18] whereas the South Africa Programme defines uptake within a given programme as the proportion initiating PrEP after both a negative test for HIV and an offer of oral PrEP.[19] In England, the PrEP Impact Trial defines uptake as the proportion initiating oral PrEP among those meeting the eligibility criteria in participating sexual health clinics.[20] A common definition of uptake has been proposed for monitoring PrEP delivery to adolescent girls and young women in sub-Saharan Africa ('the number who initiate oral PrEP among those offered PrEP')[21] and consistent definitions for the outcome measures of the PCC would be beneficial for comparing the relative success of PrEP provision programmes.

### Health inequity and PrEP

These outcome measures are important for evaluating the overall success of PrEP programmes, identifying populations with low engagement, and ensuring that programmes do not increase health inequity among at-risk populations. The latter can be achieved, in part, by considering health equity characteristics when evaluating PrEP provision programmes. Structural and environmental factors that contribute to discrepancies in HIV rates have been shown to affect HIV treatment uptake and adherence,[22 23] uptake of HIV prevention strategies in general[23] and for PrEP specifically. For example, lower levels of education, anticipated stigma,

and higher PrEP cost have been associated with lower willingness to take PrEP.[24] Low adherence to PrEP has been associated with competing survival needs (eg, financial instability, housing)[25] and substance use.[26] It is likely that other structural, social and environmental factors influence engagement at each stage of the PCC, such as those summarised by the PROGRESS-Plus framework.[27 28]

The PROGRESS-Plus framework describes a sample of characteristics that drive disparities in health.[28 29] The acronym PROGRESS refers to Place of residence (eg, rural/urban/inner city), Race/ethnicity/culture/ language, Occupation, Gender/sex, Religion, Education, Socioeconomic status and Social capital. The 'Plus' suffix includes additional categories that can attract discrimination for consideration: personal characteristics (such as age or having a disability), features of a relationship (such as being excluded from school or having unpaid care responsibilities) and time dependent relationships (such as having recently been hospitalised or receiving respite care).

Currently, the extent to which characteristics influencing health equity, such as those in the PROGRESS-Plus framework, are being investigated in outcome measures of the PCC is not well characterised. The evaluation of health equity within PrEP implementation programmes could be improved by collating current approaches for evaluating outcome measures of the PCC and existing approaches to considering health equity within them.

In order to minimise heterogeneity, this review will focus on outcome measures of the PCC in high-income countries. Although factors influencing health equity in high-income countries are likely to be similar to those in low-income and middle-income countries, how PrEP programmes are monitored, and the way vulnerable populations experience health inequity, may differ between these settings.

### Aim

This paper is a protocol for a systematic review that aims to identify and collate definitions for outcome measures of the PCC and describe how, and the extent to which, characteristics that influence health equity are taken into account in these measures in high income countries.

### Objectives

The objectives of this proposed systematic review are:
► To identify and collate definitions of outcome measures used to evaluate the main stages of the PCC (awareness, acceptability, uptake, adherence and retention) for real-world settings in high-income countries.
► To explore whether and how studies reporting outcome measures of the PCC consider the effects of health equity characteristics when reporting PrEP awareness, acceptability, uptake, adherence and retention.

► To describe the sources of data and methods used for capturing equity characteristics (eg, self-reported survey data, administrative data, linked datasets).

This review aims to describe how outcome measures for the PCC are defined and to describe how health equity characteristics are currently taken into account for these measures. This review is not intended to summarise the effectiveness of PrEP programmes at reducing health inequity or population vulnerability.

## METHOD DESIGN
### Patient and public involvement
This protocol was informed by input from a patient and public involvement (PPI) representative. A one-off PPI session was held to discuss the acceptability and accessibility of investigating equity considerations in the PCC and the appropriateness of using the PROGRESS-Plus framework to do so.

### Protocol and registration
This protocol was reported according to the Preferred Reporting Items for Systematic Review and Meta-Analysis Protocols 2015 statement[30] (online supplemental appendix 1, online supplemental file 3) and has been registered with the PROSPERO international prospective register of systematic reviews (available at: https://www.crd.york.ac.uk/prospero/display_record.php?RecordID=169779).

The review will be conducted and reported in accordance with the Preferred Reporting Items for Systematic Reviews and Meta-Analyses (PRISMA) statement guidelines[31] and the 2012 Preferred Reporting Items for Systematic Reviews and Meta-Analyses Equity extension (PRISMA-E).[32]

### Eligibility criteria
To be included in the review, papers will have to meet the following population, intervention, context, outcomes and study design elements.

### Population
Inclusion criteria: people resident in high-income countries with a national PrEP provision programme (ie, where the national governing body has approved PrEP for use and it is available from healthcare facilities and/or bespoke initiatives) or demonstration/implementation project, regardless of sexual orientation/identity, age, gender identity or occupation.

Exclusion criteria: people resident in countries without a national PrEP provision programme or demonstration/implementation project; people resident in low-income or middle-income countries.

### Intervention
The issue to be reviewed is outcome measures for five stages of the PCC (awareness, acceptability, uptake, adherence and retention). Definitions of outcome measures for the PCC in real-world settings will be collated and reviewed,

rather than the reported outcomes. For example, in studies reporting PrEP awareness, the definition of awareness and the metrics used to determine awareness are relevant to this review, whereas the reported level of awareness is not. As such, studies which include descriptions of how the main stages of the PCC will be evaluated but do not report these outcomes (eg, study protocols) will be included.

Inclusion criteria: studies reporting an outcome measure definition for at least one stage of the PCC in a real-world setting.

Exclusion criteria: studies that do not report an outcome measure definition for the PCC, randomised controlled trials (RCTs).

### Context
Inclusion criteria: studies conducted in high-income countries, as defined by the World Bank country classifications.[33] Inequity is a relative concept, and inequities experienced by vulnerable populations in lower and middle-income countries may differ from that experienced by vulnerable populations in high-income countries with well-resourced healthcare settings.

Exclusion criteria: studies conducted in low-income or middle-income countries.

### Outcomes
Primary outcomes: studies will be included if they report at least one outcome measure definition for the PCC relating to PrEP provision in a real-world setting, that is, an implementation trial, demonstration project or national programme. Studies only reporting absolute numbers for stages of the PCC will not be included.

Secondary outcomes: data on whether factors influencing health equity (defined by the PROGRESS-Plus framework) are considered for the description of baseline characteristics of participants and/or when reporting measurable outcomes for the PCC.

### Study design
Inclusion criteria: quantitative research studies including non-experimental observational studies (eg, cohort, cross-sectional and longitudinal studies) and abstracts from key international HIV/STI conferences (eg, HIV/STI World Congress, International AIDS society, Conference on Retroviruses and Opportunistic Infections). Mixed method studies will be included providing a quantified outcome measure of the PCC is reported. Studies resulting from demonstration and/or implementation trials will also be included. Study protocols, policy frameworks and clinical guidelines will be included if they include definitions of outcome measures in relation to a real-world programme; for example, definitions for how the numerator (eg, the number of people with high adherence to PrEP) and denominator (eg, the number of people taking PrEP) will be defined in a measure of PrEP adherence.

Exclusion criteria: qualitative studies, review articles, case studies, studies not involving oral emtricitabine/tenofovir

disoproxil fumarate as PrEP, and commentary or opinion pieces that do not contain a definition for an outcome measure of the PCC. RCTs will also be excluded, as RCTs in high-income countries predominately focus on safety and efficacy, rather than outcome measures of the PCC.

## Information sources
### Electronic searches
To identify studies, the following electronic databases will be searched: MEDLINE (via Ovid), EMBASE (via Ovid), PubMed, Cumulative Index to Nursing and Allied Health Literature (via EBSCO Host) and Applied Social Sciences Index and Abstracts (via ProQuest).

### Search strategy
Restrictions on English language articles as well as for articles published between 1 January 2012 (the year the US Federal Drug Administration approved Truvada (emtricitabine/tenofovir disoproxil) fumarate for PrEP use[34]) and 3 March 2020 will be applied. The searches will be re-run prior to the final analyses and further studies retrieved for inclusion.

Medical Subject Headings, subject headings and keyword lists will be created by using language that describes (i) oral PrEP and (ii) the following outcome measures of the PCC: awareness, acceptability, uptake, adherence and retention. Boolean combinations will be employed to refine the search strategy. Several key articles will be identified in the preliminary searches and used to test the sensitivity of the search strategy.

The search strategy that will be used to search MEDLINE (via the Ovid platform) is presented in online supplemental appendix 2. Search terms will be modified for databases where the indexing of subject headings differs from the terms used in MEDLINE.

The references of articles meeting the inclusion criteria will be searched to identify any relevant papers missed in the electronic searches.

## Study selection
All citations retrieved from the electronic searches will be imported into an EndNote database. After duplicates are removed, the remaining studies will be screened by title and abstract by one reviewer, with a random sample independently reviewed by a second reviewer. Reviewers will compare their decisions throughout the process and discuss all references where there was a disagreement. A third reviewer will be asked to adjudicate where discrepancies cannot be resolved. Studies remaining after the initial title and abstract screen will undergo full text screening. Following the full-text screening, data will be extracted from all studies that meet the inclusion criteria. A flow diagram will be produced to show selection process, according to PRISMA guidelines,[35] with explanation for those excluded at each stage.

## Data extraction and management
Data will be extracted into a pilot-tested, standardised form by one reviewer with a random sample checked by a second reviewer. Any discrepancies will be resolved by discussion or adjudication by a third reviewer where necessary.

All studies that meet the inclusion criteria will be described in terms of:
► Bibliographic information (first author, year of study, country of study).
► Study aims and design (including study population, sample size and data collection methods).
► Participants, demographic characteristics and number of baseline PROGRESS-Plus equity characteristics reported
► Outcome measure definitions (for PrEP awareness, acceptability, uptake, adherence and retention).
► Whether the study considered PROGRESS-Plus equity characteristics when measuring outcomes of the PCC.

For studies that consider PROGRESS-Plus equity characteristics in outcome measures, further data extraction detailing how the PROGRESS-Plus characteristics are expressed, and additional information on the study methods and study results will be extracted into an expanded data collection form.

## Risk of bias assessment of included studies
The quality of each paper included in the data extraction stage will be assessed independently by two reviewers, using the Effective Public Health Practice Project 'Quality Assessment Tool for Quantitative Studies'.[36] Individual studies will be categorised as 'strong', 'moderate' or 'weak' in the following domains: selection bias, study design, confounders, blinding, data collection method, and withdrawals and drop outs. Studies will also be given a global score to indicate overall quality. Any discrepancies between the two reviewers will be resolved by consensus, or by a third reviewer if necessary.

## Data synthesis and analysis
A narrative synthesis of results will be conducted. Individual study characteristics and outcomes will be summarised and presented in an evidence table.

To assess the extent to which studies consider the effects of PROGRESS-Plus equity characteristics in the outcome measures of the PCC, a two-stage analysis will be carried out. At the first stage, data will be presented on PCC outcome measure definitions and whether the study considers any of the PROGRESS-Plus equity characteristics for the description of baseline characteristics of study participants. The second stage we will describe whether the study considered PROGRESS-Plus characteristics in outcome measures of the PrEP continuum, and which methods were used (eg, consideration as confounders by adjusting in multivariable analysis, as effect modifiers in stratified analysis, using interaction terms in multivariable analysis, etc). The ways in which inequities are expressed (ie, whether absolute or relative) will also be reported, as the decision about whether to measure inequity in relative or absolute terms may influence their interpretation.

## DISCUSSION
To the best of the author's knowledge, this is the first systematic review to collate definitions for outcome measures of the PCC and describe the extent to (and ways in which) equity

characteristics are considered in these outcomes. The findings will have relevance to healthcare professionals, policymakers and commissioners in informing how to evaluate PrEP implementation programmes, and how to view these evaluations through an equity lens to optimise HIV prevention strategies and inform new implementation strategies for vulnerable and less advantaged populations.

## ETHICS AND DISSEMINATION

Ethical approval is not required for this study. When complete, the review results will be submitted for publication in a peer-reviewed journal and submitted to be presented at national and international conferences (where eligible). The findings, dissemination strategy, and lay summary will be discussed with a PPI panel.

**Author affiliations**
[1]Research Department of Primary Care and Population Health, University College London, London, UK
[2]Institute for Global Health, University College London, London, UK
[3]National Institute for Health Research Health Protection Research Unit in Blood Borne and Sexually Transmitted Infections at University College London, University College London, London, UK
[4]UCL Centre for Clinical Research in Infection and Sexual Health, Institute for Global Health, University College London, London, UK
[5]Blood Safety, Hepatitis, Sexually Transmitted Infections (STI) and HIV Division, National Infection Service, Public Health England, London, UK
[6]Research Department of Infection and Population Health, Institute of Global Health, University College London, London, UK

**Contributors**  MC and LKM designed the study. MC developed and refined the study protocol with contributions from all coauthors (DS, JS, GR, HM and LKM). MC prepared the manuscript. MC will undertake data collection (literature search, data extraction), analysis, interpretation and report writing. All coinvestigators will contribute to the design, analysis, interpretation and report writing. All authors read and approved the final manuscript. This protocol was informed by input from a patient and public involvement (PPI) representative. A one-off PPI session was held to discuss the acceptability and accessibility of investigating equity considerations in the PCC and the appropriateness of using the PROGRESS-Plus framework to do so. The authors would like to thank the PPI representative for their contributions to the research question and study design.

**Funding**  MC is funded by the Medical Research Council Doctoral Training Partnership (grant number: MR/N013867/1). The research is funded by the National Institute for Health Research Health Protection Research Unit (NIHR HPRU) in Blood Borne and Sexually Transmitted Infections at University College London in partnership with Public Health England (reference number: NIHR200911). The views expressed in this publication are those of the authors and not necessarily those of the NHS, the Medical Research Council, the National Institute for Health Research, the Department of Health or Public Health England.

**Competing interests**  None declared.

**Patient consent for publication**  Not required.

**Provenance and peer review**  Not commissioned; externally peer reviewed.

**ORCID iD**
Melissa Cabecinha http://orcid.org/0000-0001-6869-4692

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
