## [Reviewer comments · BMJ Open]

ARTICLE DETAILS

TITLE (PROVISIONAL)	Equity considerations in outcome measures of the HIV Pre-Exposure prophylaxis care continuum in high income countries: a systematic review protocol
AUTHORS	Cabecinha, Melissa; Solomon, Danielle; Rait, Greta; Saunders, John; Mohammed, Hamish; McDonagh, Lorraine

VERSION 1 – REVIEW

REVIEWER	Susan McAllister University of Otago
REVIEW RETURNED	22-Jun-2020

GENERAL COMMENTS	PrEP is an important intervention in the control of HIV and ensuring it is available and accessed by minority groups is vital. The authors have provided good examples of where definitions differ for outcome measures. This study will provide valuable information to clarify definitions going forward for consistency and to ensure equity issues are considered and measured in ways that are reliable. I have only two questions for consideration: 1. You say that you will include "real-world" settings and not include RCTs. If your main interest is the definitions (not results), why not include RCTs particularly RCTs that investigate adherence and retention?2. In the 4th bullet point on page 9, you say the studies will be described on outcome measure definitions and reported outcomes. Earlier you say the review is not intended to summarise the effectiveness of PrEP programmes therefore I wonder why you will report outcomes. This could be made clearer in the manuscript.
--

REVIEWER	Mehri McKellar Duke University, U.S.
REVIEW RETURNED	25-Aug-2020

GENERAL COMMENTS	I look forward to this review. I have some minor suggestions for you. I listed them by the headers used in the paper. Measurable Outcomes for PrEP Implementation: -should 'Veterans' be capitalized?-why did you capitalize 'the' and 'program in "The South Africa Program"? Health Inequity and PrEP
---

	First line - "These measures"... would say "These outcome measure" Last line - page 7 - what do you mean by 'leaving hospital or respite care'. I'm not sure if I understand this in terms of driving disparity. Do you mean that someone may have been recently hospitalized? Eligibility criteria 1. Population Can you define what a national PrEP provision program is? In the U.S., most of PrEP is delivered via private insurance (not via a national program). 3. Context Is the only reason that you are focusing on high income countries due to the difference in inequities? Seems like there is also a lack of studies in middle-lower income, although that shouldn't necessarily stop a review. 4. Outcomes Primary outcomes: What about including observational PrEP studies that include definitions of outcomes? You mention this in the Inclusion criteria. Search strategy Is the 'present' 2020? Study selection Are these author initials needed here? Somewhat distracting. Data extraction and management In the bullet items - just to be clear, you are going to report outcomes? Up to this point i thought you were just going to describe outcome measure definitions.
--	--

REVIEWER	Justin C. Smith Positive Impact Health Centers, Atlanta, GA USA Emory University Rollins School of Public Health, Atlanta, GA USA
REVIEW RETURNED	25-Aug-2020

GENERAL COMMENTS	The authors provide a very well-written overview of their proposed systematic review on a topic that will fill a critical knowledge gap in our understanding of PrEP implementation in high-income countries. There are two key places where additional specificity could strengthen the protocol; however, these do not warrant significant revisions but should be clarified in the protocol as well as the future publications that emerge from this study. 1) The authors should provide further explanation of their decision to limit PrEP studies to those that review Truvada as PrEP, given that other forms of oral PrEP (Descovy) are now available in the market. Additionally, given the progress on injectable PrEP, which as of this writing has not been approved in the US, but may receive approval during the study period, the authors should clearly state the exclusion of studies about injectable PrEP as a limitation in their discussion of the findings.
--

	2) The authors should provide a clear definition of what is meant by a "national PrEP program" in their description of inclusion/exclusion criteria. I look forward to the results of this important review.
--	--

REVIEWER	H. Nina Kim University of Washington, USA
REVIEW RETURNED	29-Sep-2020

GENERAL COMMENTS	Cebecinha and colleagues propose to conduct a systematic review to identify and collate definitions of outcome metrics for HIV Pre-exposure prophylaxis care continuum in the literature, to describe data sources/methods and to assess if these studies are considering health equity characteristics in their reports (and if so, how). Their planned approach is appropriate and clearly laid out. I had a few minor comments: (1) Study selection - unclear whether you can gain all the information you need for deciding study inclusion based off of the title & abstract alone. It would seem to me that another decision point may have to be made after full text review. (2) PROGRESS plus equity framework - should include a reference/citation for this in your proposal. (3) Worth considering further what counts as "real-world" - some might argue that implementation trials or any trial for that matter is not truly "real world" given the various exclusion criteria involved when selecting participants.
--

VERSION 1 – AUTHOR RESPONSE

Reviewer: 1 (Susan McAllister; University of Otago, Preventive and Social Medicine)

Reviewer competing interests: None declared

Comments to the Author

1. You say that you will include "real-world" settings and not include RCTs. If your main interest is the definitions (not results), why not include RCTs particularly RCTs that investigate adherence and retention?

-Response-

The reviewer is correct in that our main interest is in the definition of outcomes measures, and not the outcomes themselves. Consistent definitions of outcome measures would be beneficial for comparing the relative success of PrEP provision programmes, therefore we are only interested in definitions that have been applied to real-world programmes, i.e. outside of the highly controlled and bespoke research environment of an RCT.

RCTs in high income settings predominately focus on safety and efficacy of PrEP. This focus of this review is on collating measures used to evaluate the effectiveness of PrEP implementation, rather

than the efficacy of PrEP itself. We have added this justification as an addition to the text (“Randomised controlled trials (RCTs) will also be excluded, as RCTs in high income countries predominately focus on safety and efficacy, rather than outcome measures of the PCC” (pg 8, line 258-260)).

Definitions of outcome measures used in RCTs may not be practical or feasible to apply to real-world settings, and those definitions that are relevant to real-world programmes will likely be adopted by the studies included in our review.

-End of response-

2. In the 4th bullet point on page 9, you say the studies will be described on outcome measure definitions and reported outcomes. Earlier you say the review is not intended to summarise the effectiveness of PrEP programmes therefore I wonder why you will report outcomes. This could be made clearer in the manuscript.

-Response-

Thank you for highlighting this inconsistency. We agree with this comment, and have removed “and reported outcomes” from the manuscript (pg. 9, line 302 – 303).

-End of response-

Reviewer: 2 (Mehri McKellar; Duke University, Infectious Diseases)

Reviewer competing interests: none declared

Comments to the Author

Measurable Outcomes for PrEP Implementation:

-should 'Veterans' be capitalized?

-Response-

Thank you for bringing this to our attention. The word “veterans” is no longer capitalized. (pg. 4 line 130)

-End of Response-

-why did you capitalize 'the' and 'program in "The South Africa Program"?

-Response-

Thank you for bringing this to our attention. The word “the” is no longer capitalized in “the South Africa Program” (pg 4, line 131).

-End of response-

Health Inequity and PrEP

First line - "These measures"... would say "These outcome measure"

-Response-

We agree that the line should read “These outcome measures”, and have amended the text accordingly (pg 5 line 141).

-End of response-

Last line - page 7 - what do you mean by 'leaving hospital or respite care'. I'm not sure if I understand this in terms of driving disparity. Do you mean that someone may have been recently hospitalized?

-Response-

The text is intended to give examples of scenarios where a person may be temporarily at a disadvantage due to time-dependent relationships. Here “leaving hospital” refers to leaving hospital after having been hospitalized, and “respite care” refers to a period of receiving respite care, as opposed to care from a normal or usual caregiver. We recognize that this may not have been clear, and have amended the text to read “such as having recently been hospitalized or receiving respite care” (page 5, line 159-160).

-End of response-

Eligibility criteria

1. Population

Can you define what a national PrEP provision program is? In the U.S., most of PrEP is delivered via private insurance (not via a national program).

-Response-

We agree that the term “national PrEP provision programme” requires clarification and note that Reviewer 3 also requested a clearer definition of this term.

By “national PrEP programme” we mean where PrEP is available in government health care facilities, through universal basic healthcare coverage (where applicable), or has been approved for use by a national governing body and is available through public or private insurance. We appreciate that health care systems differ between countries, and so the term “national PrEP programme” may not accurately capture all of the provision programmes that will be included in this review. We have therefore amended the text to say “People resident in high-income countries with a national PrEP provision programme (i.e., where the national governing body has approved PrEP for use and it is available from health care facilities and/or bespoke initiatives)...” (page 6, line 209-211).

-End of response-

3. Context

Is the only reason that you are focusing on high income countries due to the difference in inequities? Seems like there is also a lack of studies in middle-lower income, although that shouldn't necessarily stop a review.

-Response-

The reviewer is correct that the reason for focusing on high-income countries is due to the differences in how inequities are experienced and expressed in high-income versus low- and middle-income countries. The inequities experienced by vulnerable populations in middle- and low- income countries may differ from those experienced by vulnerable populations with well-resourced healthcare systems. This may make it more difficult to compare how inequities are expressed between studies. Therefore we have only included high-income countries in an effort to minimize heterogeneity. Although the

authors feel that it is important to explore equity considerations in the PrEP Care Continuum in low and middle-income countries, it is beyond the scope of this review.

-End of response-

4. Outcomes

Primary outcomes:

What about including observational PrEP studies that include definitions of outcomes? You mention this in the Inclusion criteria.

-Response-

Thank you for your comment. Upon reflection, we feel that the description of included study types in the “Outcomes: Primary outcomes” section would be more appropriate for the “Study Design” section. The text “Study protocols, policy frameworks and clinical guidelines will be included if they include definitions of outcome measures in relation to a real-world program; for example definitions for how the numerator (e.g. the number of people with high adherence to PrEP) and denominator (e.g. the number of people taking PrEP) will be defined in a measure of PrEP adherence” has been moved from the “Outcomes: Primary outcomes” section to the “Study Design: Inclusions” section (page 7-8, line 251-255).

-End of response-

Search strategy

Is the 'present' 2020?

-Response-

The “present” refers to when the protocol was submitted. The actual upper date boundary used in the search strategy will be stated in full once the review is complete and the results are submitted for publication.

-End of response-

Study selection

Are these author initials needed here? Somewhat distracting.

-Response-

We agree that the author initials are not necessary in this section, and have removed them (page 8, line 282 – 291)

-End of response-

Data extraction and management

In the bullet items - just to be clear, you are going to report outcomes? Up to this point i thought you were just going to describe outcome measure definitions.

-Response-

Thank you for bringing this to our attention. The review will only describe outcome measure definitions and will not report outcomes. The text has been amended to reflect this (page 9, line 302-303)

-End of response-

Reviewer: 3 (Justin Smith; Positive Impact Health Centers)

Reviewer competing interests: None declared

Comments to the Author

1) The authors should provide further explanation of their decision to limit PrEP studies to those that review Truvada as PrEP, given that other forms of oral PrEP (Descovy) are now available in the market. Additionally, given the progress on injectable PrEP, which as of this writing has not been approved in the US, but may receive approval during the study period, the authors should clearly state the exclusion of studies about injectable PrEP as a limitation in their discussion of the findings.

-Response-

Thank you for your encouraging comments.

Currently, Descovy is not indicated in individuals at risk of HIV from receptive vaginal sex, as effectiveness in this population has not been evaluated. While this may not have an impact on the outcome measure definitions used to evaluate implementation of Descovy, it introduces a bias in terms of the populations and equity characteristics that may be included when reporting outcomes measures from the PCC for Descovy. Comparing the similarities and differences in outcome measures and reporting of equity characteristics between oral emtricitabine/tenofovir disoproxil fumarate and Descovy is beyond the scope of this review.

Regarding the exclusion of injectable PrEP, we agree that alternate PrEP delivery systems are an important consideration, particularly when viewing PrEP through an equity lens. We confirm that the exclusion of studies about injectable PrEP, and other PrEP delivery systems, will be clearly stated and discussed as a limitation in any publications and/or dissemination resulting from this review.

-End of response-

2) The authors should provide a clear definition of what is meant by a "national PrEP program" in their description of inclusion/exclusion criteria.

-Response-

We appreciate the reviewer's comment, and note that Reviewer 2 also requested a clarification. By "national PrEP programme" we mean where PrEP is available in government health care facilities, through universal basic healthcare coverage (where applicable), or has been approved for use by a national governing body and is available through public or private insurance. We appreciate that health care systems differ between countries, and so the term "national PrEP programme" may not accurately capture all of the provision programmes that will be included in this review. We have therefore amended the text to say "People resident in high-income countries with a national PrEP provision programme (i.e., where the national governing body has approved PrEP for use and it is available from health care facilities and/or bespoke initiatives)..." (page 6, line 209-211).

-End of response-

I look forward to the results of this important review.

Reviewer: 4 (Nina Kim; University of Washington)

Reviewer competing interests: None

Comments to the Author

Cebecinha and colleagues propose to conduct a systematic review to identify and collate definitions of outcome metrics for HIV Pre-exposure prophylaxis care continuum in the literature, to describe data sources/methods and to assess if these studies are considering health equity characteristics in their reports (and if so, how). Their planned approach is appropriate and clearly laid out. I had a few minor comments:

(1) Study selection - unclear whether you can gain all the information you need for deciding study inclusion based off of the title & abstract alone. It would seem to me that another decision point may have to be made after full text review.

-Response-

Thank you for your comment. In the text we state "Following the full-text screening, data will be extracted from all studies that meet the inclusion criteria", however this was included in the "Data Extraction and Management" section. We agree that this would be more appropriate in the "Study Selection" section, and have moved it accordingly (page 8, line 287-288)

-End of response-

(2) PROGRESS plus equity framework - should include a reference/citation for this in your proposal.

-Response-

We would like to draw the reviewer's attention to page 5 line 154, which is the first mention of the PROGRESS-Plus framework in the main body of the text. Two citations have been included for the framework in this instance. The citations were not carried forward for subsequent mentions of the framework in the text.

-End of response-

(3) Worth considering further what counts as "real-world" - some might argue that implementation trials or any trial for that matter is not truly "real world" given the various exclusion criteria involved when selecting participants.

-Response-

We acknowledge the reviewer's point that implementation trials may not be considered to be truly "real-world" settings by some, however for the purposes of this review, we feel that including implementation trials in our definition of "real-world" settings is appropriate. For this review we are interested in the definitions used for outcomes measures of the PrEP care continuum, rather than the outcomes themselves. PrEP implementation trials generally include outcomes on the appropriateness and feasibility of a programme either within current healthcare infrastructure, or as bespoke initiatives that may be adopted as standard "real-world" settings.

-End of response-